# Discovering Neural Wirings

**Mitchell Wortsman**[1,2], **Ali Farhadi**[1,2,3], **Mohammad Rastegari**[1,3]
[1]PRIOR @ Allen Institute for AI, [2]University of Washington, [3]XNOR.AI
mitchnw@cs.washington.edu, {ali, mohammad}@xnor.ai

## Abstract

The success of neural networks has driven a shift in focus from feature engineering to architecture engineering. However, successful networks today are constructed using a small and manually defined set of building blocks. Even in methods of neural architecture search (NAS) the network connectivity patterns are largely constrained. In this work we propose a method for discovering neural wirings. We relax the typical notion of layers and instead enable channels to form connections independent of each other. This allows for a much larger space of possible networks. The wiring of our network is not fixed during training – as we learn the network parameters we also learn the structure itself. Our experiments demonstrate that our learned connectivity outperforms hand engineered and randomly wired networks. By learning the connectivity of MobileNetV1 [12] we boost the ImageNet accuracy by $10\%$ at $\sim 41$M FLOPs. Moreover, we show that our method generalizes to recurrent and continuous time networks. Our work may also be regarded as unifying core aspects of the neural architecture search problem with sparse neural network learning. As NAS becomes more fine grained, finding a good architecture is akin to finding a sparse subnetwork of the complete graph. Accordingly, DNW provides an effective mechanism for discovering sparse subnetworks of predefined architectures in a single training run. Though we only ever use a small percentage of the weights during the forward pass, we still play the so-called initialization lottery [8] with a combinatorial number of subnetworks. Code and pretrained models are available at https://github.com/allenai/dnw while additional visualizations may be found at https://mitchellnw.github.io/blog/2019/dnw/.

## 1   Introduction

Deep neural networks have shifted the prevailing paradigm from *feature engineering* to *feature learning*. The architecture of deep neural networks, however, must still be hand designed in a process known as *architecture engineering*. A myriad of recent efforts attempt to automate the process of the architecture design by searching among a set of smaller well-known building blocks [30, 34, 37, 19, 2, 20]. While methods of search range from reinforcement learning to gradient based approaches [34, 20], the space of possible connectivity patterns is still largely constrained. NAS methods explore wirings between predefined blocks, and [28] learns the recurrent structure of CNNs. We believe that more efficient solutions may arrive from searching the space of wirings at a more fine grained level, i.e. single channels.

In this work, we consider an unconstrained set of possible wirings by allowing channels to form connections independent of each other. This enables us to discover a wide variety of operations (e.g. depthwise separable convs [12], channel shuffle and split [36], and more). Formally, we treat the network as a large *neural graph* where each each node processes a single channel.

One key challenge lies in searching the space of all possible wirings – the number of possible sub-graphs is combinatorial in nature. When considering thousands of nodes, traditional search methods are either prohibitive or offer approximate solutions. In this paper we introduce a simple

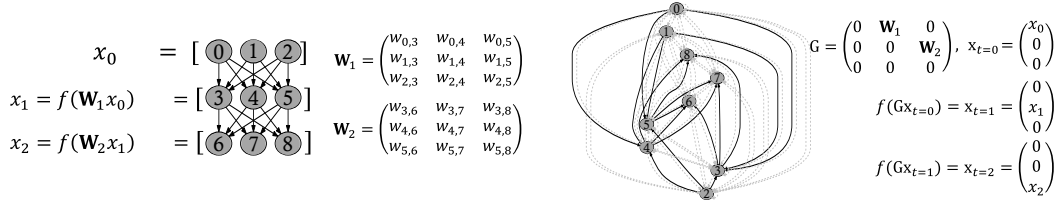

Figure 1: **Dynamic Neural Graph:** A 3-layer perceptron (*left*) can be expressed by a dynamic neural graph with 3 time steps (*right*).

and efficient algorithm for discovering neural wirings (DNW). Our method searches the space of all possible wirings with a simple modification of the backwards pass.

Recent work in randomly wired neural networks [35] aims to explore the space of novel neural network wirings. Intriguingly, they show that constructing neural networks with random graph algorithms often outperforms a manually engineered architecture. However, these wirings are fixed at training.

Our method for discovering neural wirings is as follows: First, we consider the sole constraint that that the total number of edges in the *neural graph* is fixed to be $k$. Initially we randomly assign a weight to each edge. We then choose the weighted edges with the highest magnitude and refer to the remaining edges as *hallucinated*. As we train, we modify the weights of *all* edges according to a specified update rule. Accordingly, a hallucinated edge may strengthen to a point it replaces a real edge. We tailor the update rule so that when *swapping* does occur, it is beneficial.

We consider the application of DNW for *static* and *dynamic* neural graphs. In the *static* regime each node has a single output and the graphical structure is acyclic. In the case of a *dymanic neural graph* we allow the state of a node to vary with time. *Dymanic neural graphs* may contain cycles and express popular sequential models such as LSTMs [11]. As *dymanic neural graphs* are strictly more expressive than *static neural graphs*, they can also express feed-forward networks (as in Figure 1).

Our work may also be regarded as a unification between the problem of neural architecture search and sparse neural network learning. As NAS becomes less restrictive and more fine grained, finding a good architecture is akin to finding a sparse sub-network of the complete graph. Accordingly, DNW provides an effective mechanism for discovering sparse networks in a single training run.

The *Lottery Ticket Hypothesis* [8, 9] demonstrates that dense feed-forward neural networks contain so-called *winning-tickets*. These *winning-tickets* are sparse subnetworks which, when reset to their initialization and trained in isolation, reach an accuracy comparable to their dense counterparts. This hypothesis articulate an advantage of overparameterization during training – having more parameters increases the chance of winning the *initialization lottery*. We leverage this idea to train a sparse neural network without retraining or fine-tuning. Though we only ever use a small percentage of the weights during the forward pass, we still play the lottery with a combinatorial number of sub-networks.

We demonstrate the efficacy of DNW on small and large scale data-sets, and for feed-forward, recurrent, continuous, and sparse networks. Notably, we augment MobileNetV1 [12] with DNW to achieve a $10\%$ improvement on ImageNet [5] from the hand engineered MobileNetV1 at $\sim 41\text{M}$ FLOPs[1].

## 2   Discovering Neural Wirings

In this section we describe our method for jointly discovering the structure and learning the parameters of a neural network. We first consider the algorithm in a familiar setting, a feed-forward neural network, which we abstract as a *static neural graph*. We then present a more expressive *dynamic neural graph* which extends to discrete and continuous time and generalizes feed-forward, recurrent, and continuous time neural networks.

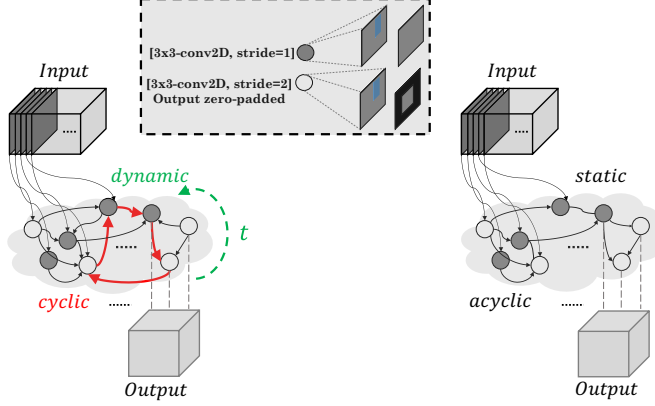

Figure 2: An example of a dynamic (left) and static (right) neural graph. Details in Section 2.3.

## 2.1 Static Neural Graph

A *static neural graph* is a directed acyclic graph $\mathcal{G} = (\mathcal{V}, \mathcal{E})$ consisting of nodes $\mathcal{V}$ and edges $\mathcal{E} \subseteq \mathcal{V} \times \mathcal{V}$. The state of a node $v \in \mathcal{V}$ is given by the random variable $Z_v$. At each node $v$ we apply a function $f_{\theta_v}$ and with each edge $(u, v)$ we associate a weight $w_{uv}$. In the case of a multi-layer perceptron, $f$ is simply a parameter-free non-linear activation like ReLU [17].

For any set $\mathcal{A} \subseteq \mathcal{V}$ we let $\mathbf{Z}_{\mathcal{A}}$ denote $(Z_v)_{v \in \mathcal{A}}$ and so $\mathbf{Z}_{\mathcal{V}}$ is the state of all nodes in the network.

$\mathcal{V}$ contains a subset of input nodes $\mathcal{V}_0$ with no parents and output nodes $\mathcal{V}_E$ with no children. The input data $\mathcal{X} \sim p_x$ flows into the network through $\mathcal{V}_0$ as $\mathbf{Z}_{\mathcal{V}_0} = g_\phi(\mathcal{X})$ for a function $g$ which may have parameters $\phi$. Similarly, the output of the network $\hat{\mathcal{Y}}$ is given by $h_\psi(\mathbf{Z}_{\mathcal{V}_E})$.

$$Z_v = \begin{cases} f_{\theta_v}\left(\sum_{(u,v) \in \mathcal{E}} w_{uv} Z_u\right) & v \in \mathcal{V} \setminus \mathcal{V}_0 \\ g_\phi^{(v)}(\mathcal{X}) & v \in \mathcal{V}_0. \end{cases} \tag{1}$$

For brevity, we let $\mathcal{I}_v$ denote the "input" to node $v$, where $\mathcal{I}_v$ may be expressed

$$\mathcal{I}_v = \sum_{(u,v) \in \mathcal{E}} w_{uv} Z_u. \tag{2}$$

In this work we consider the case where the input and output of each node is a two-dimensional matrix, commonly referred to as a channel. Each node performs a non-linear activation followed by normalization and convolution (which may be strided to reduce the spatial resolution). As in [35], we no longer conform to the traditional notion of "layers" in a deep network.

The combination of a separate $3 \times 3$ convolution for each channel (depthwise convolution) followed by a $1 \times 1$ convolution (pointwise convolution) is often referred to as a depthwise seperable convolution, and is essential in efficient network design [12, 22]. With a *static neural graph* this process may be interpreted equivalently as a $3 \times 3$ convolution at each node followed by information flow on a complete bipartite graph.

## 2.2 Discovering a $k$-Edge neural graph

We now outline our method for discovering the edges of a static neural graph subject to the constraint that the total number of edges must not exceed $k$.

We consider a set of real edges $\mathcal{E}$ and a set of *hallucinated* edges $\mathcal{E}_{\text{hal}} = \mathcal{V} \times \mathcal{V} \setminus \mathcal{E}$. The real edge set is comprised of the $k$-edges which have the largest magnitude weight. As we allow the magnitude of the weights in both sets to change throughout training the edges in $\mathcal{E}_{\text{hal}}$ may replace those in $\mathcal{E}$.

Consider a *hallucinated* edge $(u, v) \notin \mathcal{E}$. If the gradient is pushing $\mathcal{I}_v$ in a direction which aligns with $Z_u$, then our update rule strengthens the magnitude of the weight $w_{uv}$. If this alignment happens consistently then $w_{uv}$ will be eventually be strong enough to enter the real edge set $\mathcal{E}$. As the total

**Algorithm 1** DNW-Train$(\mathcal{V}, \mathcal{V}_0, \mathcal{V}_E, g_\phi, h_\psi, \{f_{\theta_v}\}_{v \in \mathcal{V}}, p_{xy}, k, \mathcal{L})$

1: **for** each pair of nodes $(u, v)$ such that $u < v$ **do** ▷ Initialize
2:     Initialize $w_{uv}$ by independently sampling from a uniform distribution.
3: **for** each training iteration **do**
4:     Sample mini batch of data and labels $(\mathcal{X}, \mathcal{Y}) = \{(\mathcal{X}_i, \mathcal{Y}_i)\}$ using $p_{xy}$ ▷ Sample data
5:     $\mathcal{E} \leftarrow \{(u, v) : |w_{uv}| \geq \tau\}$ where $\tau$ is chosen so that $|\mathcal{E}| = k$ ▷ Choose edges
6:     $Z_v \leftarrow \begin{cases} f_{\theta_v}\left( \sum_{(u,v) \in \mathcal{E}} w_{uv} Z_u \right) & v \in \mathcal{V} \setminus \mathcal{V}_0 \\ g_\phi^{(v)}(\mathcal{X}) & v \in \mathcal{V}_0 \end{cases}$ ▷ Forward pass
7:     $\hat{\mathcal{Y}} = h_\psi\left( \{Z_v\}_{v \in \mathcal{V}_E} \right)$ ▷ Compute output
8:     Update $\phi, \{\theta_v\}_{v \in \mathcal{V}}, \psi$ via SGD & Backprop [26] using loss $\mathcal{L}\left( \hat{\mathcal{Y}}, \mathcal{Y} \right)$
9:     **for** each pair of nodes $(u, v)$ such that $u < v$ **do** ▷ Update edge weights
10:         $w_{uv} \leftarrow w_{uv} + \left\langle Z_u, -\alpha \frac{\partial \mathcal{L}}{\partial \mathcal{I}_v} \right\rangle$ ▷ Recall $\mathcal{I}_v = \sum_{(u,v) \in \mathcal{E}} w_{uv} Z_u$

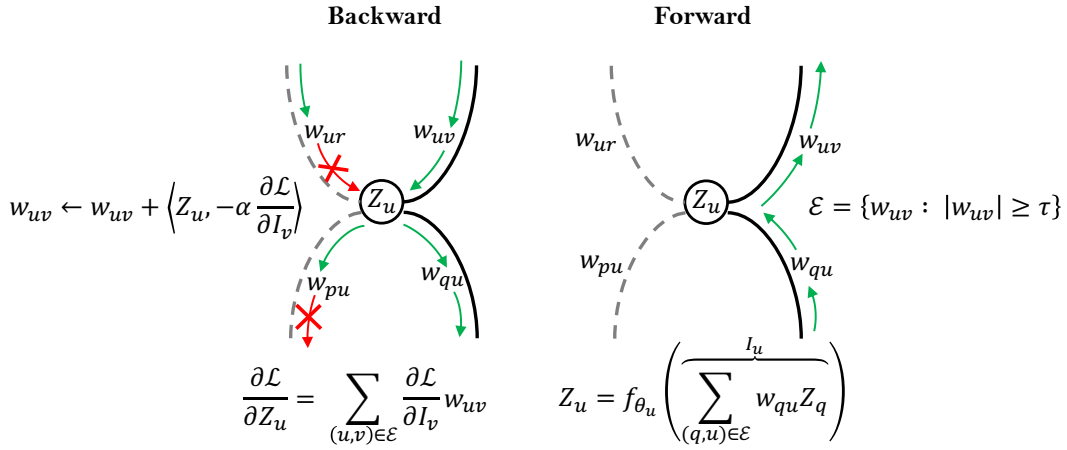

Figure 3: **Gradient flow:** On the forward pass we use only on the *real* edges. On the backwards pass we allow the gradient to flow *to* but not *through* the hallucinated edges (as in Algorithm 1).

number of edges is conserved, when $(u, v)$ enters the edge set $\mathcal{E}$ another edge is removed and placed in $\mathcal{E}_{\text{hal}}$. This procedure is detailed by Algorithm 1, where $\mathcal{V}$ is the node set, $\mathcal{V}_0, \mathcal{V}_E$ are the input and output node sets, $g_\phi, h_\psi$ and $\{f_{\theta_v}\}_{v \in \mathcal{V}}$ are the input, output, and node functions, $p_{xy}$ is the data distribution, $k$ is the number of edges in the graph and $\mathcal{L}$ is the loss.

In practice we may also include a momentum and weight decay[2] term in the weight update rule (line 10 in Algorithm 1). In fact, the weight update rule looks nearly identical to that in traditional SGD & Backprop but for one key difference: we **allow** the gradient to flow *to* edges which did not exist during the forward pass. Importantly, we **do not allow** the gradient to flow *through* these edges and so the rest of the parameters update as in traditional SGD & Backprop. This gradient flow is illustrated in Figure 3.

Under certain conditions we formally show that swapping an edge from $\mathcal{E}_{\text{hal}}$ to $\mathcal{E}$ decreases the loss $\mathcal{L}$. We first consider the simple case where the *hallucinated edge* $(i, k)$ replaces $(j, k) \in \mathcal{E}$. In Section C we discuss the proof to a more general case.

We let $\tilde{w}$ to denote the weight $w$ after the weight update rule $\tilde{w}_{uv} = w_{uv} + \left\langle Z_u, -\alpha \frac{\partial \mathcal{L}}{\partial \mathcal{I}_v} \right\rangle$. We assume that $\alpha$ is small enough so that $\text{sign}(\tilde{w}) = \text{sign}(w)$.

**Claim:** Assume $\mathcal{L}$ is Lipschitz continuous. There exists a learning rate $\alpha^* > 0$ such that for $\alpha \in (0, \alpha^*)$ the process of swapping $(i, k)$ for $(j, k)$ will decrease the loss on the mini-batch when the state of the nodes are fixed and $|w_{ik}| < |w_{jk}|$ but $|\tilde{w}_{ik}| > |\tilde{w}_{jk}|$.

*Proof.* Let $\mathcal{A}$ be value of $\mathcal{I}_k$ after the update rule if $(j,k)$ is replaced with $(i,k)$. Let $\mathcal{B}$ be the state of $\mathcal{I}_k$ after the update rule if we do not allow for swapping. $\mathcal{A}$ and $\mathcal{B}$ are then given by

$$\mathcal{A} = \tilde{w}_{ik}Z_i + \sum_{(u,k)\in\mathcal{E},\ u\neq i,j} \tilde{w}_{uk}Z_u, \qquad \mathcal{B} = \tilde{w}_{jk}Z_j + \sum_{(u,k)\in\mathcal{E},\ u\neq i,j} \tilde{w}_{uk}Z_u. \tag{3}$$

Additionally, let $g = -\alpha\frac{\partial\mathcal{L}}{\partial\mathcal{I}_k}$ be the direction in which the loss most steeply descends with respect to $\mathcal{I}_k$. By *Lemma 1* (Section D of the Appendix) it suffices to show that moving $\mathcal{I}_k$ towards $\mathcal{A}$ is more aligned with $g$ then moving $\mathcal{I}_k$ towards $\mathcal{B}$. Formally we wish to show that

$$\langle \mathcal{A} - \mathcal{I}_k, g \rangle \geq \langle \mathcal{B} - \mathcal{I}_k, g \rangle \tag{4}$$

which simplifies to

$$\tilde{w}_{ik}\langle Z_i, g\rangle \geq \tilde{w}_{jk}\langle Z_j, g\rangle \tag{5}$$
$$\iff \tilde{w}_{ik}(\tilde{w}_{ik} - w_{ik}) \geq \tilde{w}_{jk}(\tilde{w}_{jk} - w_{jk}). \tag{6}$$

In the case where $\tilde{w}_{ik}$ and $(\tilde{w}_{ik}-w_{ik})$ have the same sign but $\tilde{w}_{jk}$ and $(\tilde{w}_{jk}-w_{jk})$ have different signs the inequality immediately holds. This corresponds to the case where $w_{ik}$ *increases* in magnitude but $w_{jk}$ *decreases* in magnitude. The opposite scenario ($w_{ik}$ *decreases* in magnitude but $w_{jk}$ *increases*) is impossible since $|w_{ik}| < |w_{jk}|$ but $|\tilde{w}_{ik}| > |\tilde{w}_{jk}|$.

We now consider the scenario where both sides of the inequality (equation 6) are positive. Simplifying further we obtain

$$(\tilde{w}_{jk}w_{jk} - \tilde{w}_{ik}w_{ik}) \geq \left(\tilde{w}_{jk}^2 - \tilde{w}_{ik}^2\right) \tag{7}$$

and are now able to identify a range for $\alpha$ such that the inequality above is satisfied. By assumption the right hand side is less than 0 and $\text{sign}(\tilde{w}) = \text{sign}(w)$ so $\tilde{w}w = |\tilde{w}||w|$. Accordingly, it suffices to show that

$$|\tilde{w}_{jk}||w_{jk}| - |\tilde{w}_{ik}||w_{ik}| \geq 0. \tag{8}$$

If we let $\epsilon = |w_{jk}| - |w_{ik}|$ and $\alpha^* = \sup\{\alpha : |\tilde{w}_{ik}| \leq |\tilde{w}_{jk}| + \epsilon|\tilde{w}_{jk}||w_{ik}|^{-1}\}$, then for $\alpha \in (0, \alpha^*)$

$$|\tilde{w}_{jk}||w_{jk}| - |\tilde{w}_{ik}||w_{ik}| \geq |\tilde{w}_{jk}|\left(\underbrace{|w_{jk}| - |w_{ik}|}_{=\epsilon} - \epsilon\right) = 0 \tag{9}$$

the inequality (equation 7) is satisfied. Here we are implicitly using our assumption that the gradient is bounded and we may "tune" $\alpha$ to control the magnitude $|w_{ik}| - |\tilde{w}_{jk}|$. In the case where $\alpha = \inf\{\alpha : |\tilde{w}_{ik}| > |\tilde{w}_{jk}|\}$ the right hand side of equation 7 becomes 0 while the left hand side is $\epsilon > 0$.

In Section E of the appendix we discuss the effect of $\theta_v$ on $w_{uv}$. In Section F of the Appendix, we show that the update rule is equivalently a straight-through estimator [1].

## 2.3 Dynamic Neural Graph

We now consider a more general setting where the state of each node $Z_v(t)$ may vary through time. We refer to this model as a *dynamic neural graph*.

The initial conditions of a *dynamic neural graph* are given by

$$Z_v(0) = \begin{cases} g_\phi^{(v)}(\mathcal{X}) & v \in \mathcal{V}_0 \\ 0 & v \in \mathcal{V} \setminus \mathcal{V}_0 \end{cases} \tag{10}$$

where $\mathcal{V}_0$ is a designated set of input nodes, which may now have parents.

**Discrete Time Dynamics:** For a discrete time neural graph we consider times $\ell \in \{0, 1, ..., L\}$. The dynamics are then given by

$$Z_v(\ell + 1) = f_{\theta_v}\left(\sum_{(u,v)\in\mathcal{E}} w_{uv}Z_u(\ell),\ \ell\right) \tag{11}$$

and the network output is $\hat{y} = h_\psi(\mathbf{Z}_{\mathcal{V}_E}(L))$. We may express equation 11 more succinctly as

$$\mathbf{Z}_{\mathcal{V}}(\ell + 1) = \mathbf{f}_\theta\left(\mathcal{A}_{\mathcal{G}}\mathbf{Z}_{\mathcal{V}}(\ell),\ \ell\right) \tag{12}$$

where $\mathbf{Z}_{\mathcal{V}}(\ell) = (Z_v(\ell))_{v \in \mathcal{V}}$, $\mathbf{f}_\theta(\mathbf{z}, \ell) = (f_{\theta_v}(z_v, \ell))_{v \in \mathcal{V}}$, and $\mathcal{A}_{\mathcal{G}}$ is the weighted adjacency matrix for graph $\mathcal{G}$. Equation 12 suggests the following interpretation: At each time step we send information through the edges using $\mathcal{A}_{\mathcal{G}}$ then apply a function at each node.

**Continuous Time Dynamics:** As in [3], we consider the case where $t$ may take on a continuous range of values. We then arrive at dynamics given by

$$\nabla \, \mathbf{Z}_{\mathcal{V}}(t) = \mathbf{f}_\theta \left( \mathcal{A}_{\mathcal{G}} \mathbf{Z}_{\mathcal{V}}(t), \, t \right). \tag{13}$$

Interestingly, if $\mathcal{V}_0$ is a strict subset of $\mathcal{V}$ we uncover an Augmented Neural ODE [7].

The discrete time case is unifying in the sense that it may also express any static neural graph. In Figure 1 we illustrate than an MLP may also be expressed by a discrete time neural graph. Additionally, the discrete time dynamics are able to capture sequential models such as LSTMs [11], as long as we allow input to flow into $\mathcal{V}_0$ at any time.

In continuous time it is not immediately obvious how to incorporate strided convolutions. One approach is to keep the same spatial resolution throughout and pad with zeros after applying strided convolutions. This design is illustrated by Figure 2.

We may also apply Algorithm 1 to learn the structure of dynamic neural graphs. One may use backpropagation through time [33] and the adjoint-sensitivity method [3] for optimization in the discrete and continuous time settings respectively. In Section 3.1, we demonstrate empirically that our method performs better than a random graph, though we do not formally justify the application of our algorithm in this setting.

## 2.4 Implementation details for Large Scale Experiments

For large scale experiments we do not consider the dynamic case as optimization is too expensive. Accordingly, we now present our method for constructing a large and efficient *static neural graph*. With this model we may jointly learn the structure of the graph along with the parameters on ImageNet [5]. As illustrated by Table 5 our model closely follows the structure of MobileNetV1 [12], and so we refer to it as MobileNetV1-DNW. We consider a separate *neural graph* for each spatial resolution – the output of graph $\mathcal{G}_i$ is the input of graph $\mathcal{G}_{i+1}$. For width multiplier [12] $d$ and spatial resolution $s \times s$ we constrain MobileNetV1-DNW to have the same number of edges for resolution $s \times s$ as the corresponding MobileNetV1 $\times d$. We use a *slightly smaller* width multiplier to obtain a model with similar FLOPs as we do not explicitly reduce the number of depthwise convolutions in MobileNetV1-DNW. However, we do find that neurons often *die* (have no output) and we may then skip the depthwise convolution during inference. Note that if we interpret a pointwise convolution with $c_1$ input channels and $c_2$ output channels as a complete bipartite graph then the number of edges is simply $c_1 * c_2$.

We also constrain the longest path in graph $\mathcal{G}$ to be equivalent to the number of layers of the corresponding MobileNetV1. We do so by partitioning the nodes $\mathcal{V}$ into blocks $\mathcal{B} = \{\mathcal{B}_0, ..., \mathcal{B}_{L-1}\}$ where $\mathcal{B}_0$ is the input nodes $\mathcal{V}_0$, $\mathcal{B}_{L-1}$ is output nodes $\mathcal{V}_E$, and we only allow edges between nodes in $\mathcal{B}_i$ and $\mathcal{B}_j$ if $i < j$. The longest path in a graph with $L$ blocks is then $L - 1$. Splitting the graph into blocks also improves efficiency as we may operate on one block at a time. The structure of MobileNetV1 may be recovered by considering a complete bipartite graph between adjacent blocks.

The operation $f_{\theta_v}$ at each non-output node is a batch-norm [14] (2 parameters), ReLU [17], $3 \times 3$ convolution (9 parameters) triplet. There are no operations at the output nodes. When the spatial resolution decreases in MobileNetV1 we change the convolutional stride of the input nodes to 2.

In models denoted MobileNetV1-DNW-Small ($\times d$) we also limit the last fully connected (FC) layer to have the same number of edges as the FC layer in MobileNetV1 ($\times d$). In the normal setting of MobileNetV1-DNW we do not modify the last FC layer.

# 3 Experiments

In this section we demonstrate the effectiveness of DNW for image classification in small and large scale settings. We begin by comparing our method with a random wiring on a small scale dataset

Table 1: Testing a tiny (41k parameters) classifier on CIFAR-10 [16] in static and dynamic settings shown as mean and standard deviation (std) over 5 runs.

| Model | Accuracy |
|---|---|
| Static (RG) | $76.1 \pm 0.5\%$ |
| Static (DNW) | $80.9 \pm 0.6\%$ |
| Discrete Time (RG) | $77.3 \pm 0.7\%$ |
| Discrete Time (DNW) | $82.3 \pm 0.6\%$ |
| Continuous (RG) | $78.5 \pm 1.2\%$ |
| Continuous (DNW) | $83.1 \pm 0.3\%$ |

Table 2: Other methods for discovering wirings (using the architecture described in Table 5) tested on CIFAR-10 shown as mean and std over 5 runs. Models with † first require the complete graph to be trained.

| Model | Accuracy |
|---|---|
| MobileNetV1 ($\times 0.25$) | $86.3 \pm 0.2\%$ |
| MobileNetV1-RG($\times 0.225$) | $87.2 \pm 0.1\%$ |
| No Update Rule | $86.7 \pm 0.5\%$ |
| L1 + Anneal | $84.3 \pm 0.6\%$ |
| TD $\rho = 0.95$ | $89.2 \pm 0.4\%$ |
| Lottery Ticket (one-shot)† | $87.9 \pm 0.3\%$ |
| Fine Tune $\alpha = 0.1$† | $89.4 \pm 0.2\%$ |
| Fine Tune $\alpha = 0.01$† | $89.7 \pm 0.1\%$ |
| Fine Tune $\alpha = 0.001$† | $88.7 \pm 0.2\%$ |
| MobileNetV1-DNW($\times 0.225$) | $89.7 \pm 0.2\%$ |

and model. This allows us to experiment in static, discrete time, and continuous settings. Next we explore the use of DNW at scale with experiments on ImageNet [5] and compare DNW with other methods of discovering network structures. Finally we use our algorithm to effectively train sparse neural networks without retraining or fine-tuning.

Throughout this section we let RG denote our primary baseline – a **randomly wired graph**. To construct a randomly wired graph with $k$-edges we assign a uniform random weight to each edge then pick the $k$ edges with the largest magnitude weights. As shown in [35], random graphs often outperform manually designed networks.

### 3.1 Small Scale Experiments For Static and Dynamic Neural Graphs

We begin by training tiny classifiers for the CIFAR-10 dataset [16]. Our initial aim is not to achieve state of the art performance but instead to explore DNW in the static, discrete, and continuous time settings. As illustrated by Table 1, our method outperforms a random graph by a large margin.

The image is first downsampled[3] then each channel is given as input to a node in a *neural graph*. The static graph uses 5 blocks and the discrete time graph uses 5 time steps. For the continuous case we backprop through the operation of an adaptive ODE solver[4]. The models have 41k parameters. At each node we perform Instance Normalization [32], ReLU, and a $3 \times 3$ single channel convolution.

### 3.2 ImageNet Classification

For large scale experiments on ImageNet [5] we are limited to exploring DNW in the static case (recurrent and continuous time networks are more expensive to optimize due to lack of parallelization). Although our network follows the simple structure of MobileNetV1 [12] we are able to achieve higher accuracy than modern networks which are more advanced and optimized. Notably, MobileNetV2 [27] extends MobileNetV1 by adding residual connections and linear bottlenecks and ShuffleNet [36, 22] introduces channel splits and channel shuffles. The results of the large scale experiments may be found in Table 3.

As standard, we have divided the results of Table 3 to consider models which have similar FLOPs. In the more sparse case ($\sim 41M$ FLOPs) we are able to use DNW to boost the performance of MobileNetV1 by $10\%$. Though random graphs perform extremely well we still observe a $7\%$ boost in performance. In each experiment we train for 250 epochs using Cosine Annealing as the learning rate scheduler with initial learning rate 0.1, as in [35]. Models using random graphs have considerably more FLOPs as nearly all depthwise convolutions must be performed. DNW allows neurons to die and we may therefore skip many operations.

Table 3: ImageNet Experiments (see Section 2.4 for more details). Models with $^*$ use the implementations of [22]. Models with multiples asterisks use different image resolutions so that the FLOPs is comparable (see Table 8 in [22] for more details).

| Model | Params | FLOPs | Accuracy |
|---|---|---|---|
| MobileNetV1 ($\times 0.25$) [12] | 0.5M | 41M | 50.6% |
| X-4 MobileNetV1 [25] | — | $> 50$M | 54.0% |
| MobileNetV2 ($\times 0.15$)$^*$ [27] | — | 39M | 44.9% |
| MobileNetV2 ($\times 0.4$)$^{**}$ | — | 43M | 56.6% |
| DenseNet ($\times 0.5$)$^*$ [13] | — | 42M | 41.1% |
| Xception ($\times 0.5$)$^*$ [4] | — | 40M | 55.1% |
| ShuffleNetV1 ($\times 0.5$, $g = 3$) [36] | — | 38M | 56.8% |
| ShuffleNetV2 ($\times 0.5$) [22] | 1.4M | 41M | 60.3% |
| MobileNetV1-RG($\times 0.225$) | 1.2M | 55.7M | 53.3% |
| MobileNetV1-DNW-Small ($\times 0.15$) | 0.24M | 22.1M | 50.3% |
| MobileNetV1-DNW-Small ($\times 0.225$) | 0.4M | 41.2M | 59.9% |
| MobileNetV1-DNW($\times 0.225$) | 1.1M | 42.1M | 60.9% |
| MnasNet-search1 [30] | 1.9M | 65M | 64.9% |
| MobileNetV1-DNW($\times 0.3$) | 1.3M | 66.7M | 65.0% |
| MobileNetV1 ($\times 0.5$) | 1.3M | 149M | 63.7% |
| MobileNetV2 ($\times 0.6$)$^*$ | — | 141M | 66.6% |
| MobileNetV2 ($\times 0.75$)$^{***}$ | — | 145M | 67.9% |
| DenseNet ($\times 1$)$^*$ | — | 142M | 54.8% |
| Xception ($\times 1$)$^*$ | — | 145M | 65.9% |
| ShuffleNetV1 ($\times 1$, $g = 3$) | — | 140M | 67.4% |
| ShuffleNetV2 ($\times 1$) | 2.3M | 146M | 69.4% |
| MobileNetV1-RG($\times 0.49$) | 1.8M | 170M | 64.1% |
| MobileNetV1-DNW($\times 0.49$) | 1.8M | 154M | 70.4% |

## 3.3 Related Methods

We compare DNW with various methods for discovering neural wirings. In Table 2 we use the structure of MobileNetV1-DNW but try other methods which find $k$-edge sub-networks. The experiments in Table 2 are conducted using CIFAR-10 [16]. We train for 160 epochs using Cosine Annealing as the learning rate scheduler with initial learning rate $\alpha = 0.1$ unless otherwise noted.

**The Lottery Ticket Hypothesis:** The authors of [8, 9] offer an intriguing hypothesis: sparse sub-networks may be trained in isolation when reset to their initialization. However, their method for finding so-called winning tickets is quite expensive as it requires training the full graph from scratch. We compare with **one-shot** pruning from [9]. One-shot pruning is more comparable in training FLOPS than iterative pruning [8], though both methods are more expensive in training FLOPS than DNW. After training the full network $\mathcal{G}_{\text{full}}$ (i.e. no edges pruned) the optimal sub-network $\mathcal{G}_k$ with $k$-edges is chosen by taking the weights with the highest magnitude. In the row denoted *Lottery Ticket* we retrain $\mathcal{G}_k$ using the initialization of $\mathcal{G}_{\text{full}}$. We also initialize $\mathcal{G}_k$ with the weights of $\mathcal{G}_{\text{full}}$ *after* training – denoted by **FT** for *fine-tune* (we try different initial learning rates $\alpha$). Though these experiments perform comparably with DNW, their training is more expensive as the full graph must initially be trained.

**Exploring Randomly Wired Networks for Image Recognition:** The authors of [35] explore "*a more diverse set of connectivity patterns through the lens of randomly wired neural networks.*" They achieve impressive performance on ImageNet [5] using random graph algorithms to generate the structure of a neural network. Their network connectivity, however, is fixed during training. Throughout this section we have a random graph (denoted **RG**) as our primary baseline – as in [35] we have seen that random graphs outperform hand-designed networks.

**No Update Rule:** In this ablation on DNW we do not apply the update rule to the hallucinated edges. An edge may only leave the hallucinated edge set if the magnitude of a real edge is sufficiently *decreased*. This experiment demonstrates the importance of the update rule.

**L1 + Anneal:** We experiment with a simple pruning technique – start with a fully connected graph and remove edges by magnitude throughout training until there are only $k$ remaining. We found that accuracy was much better if we added an L1 regularization term.

Table 4: Training a tuned version of ResNet50 on ImageNet with modern optimization techniques, as in Appendix C of [6]. For *All Layers Sparse*, every layer has a fixed sparsity. In contrast, we leave the very first convolution dense for *First Layer Dense*. The parameters in the first layer constitute only 0.04% of the total network.

| Method | Weights (%) | Top-1 Accuracy | Top-5 Accuracy |
|---|---|---|---|
| Sparse Networks from Scratch [6] | 10% | 72.9% | 91.5% |
| Ours - All Layers Sparse | 10% | 74.0% | 92.0% |
| Ours - First Layer Dense | 10% | 75.0% | 92.5% |
| Sparse Networks from Scratch [6] | 20% | 74.9% | 92.5% |
| Ours - All Layers Sparse | 20% | 76.2% | 93.0% |
| Ours - First Layer Dense | 20% | 76.6% | 93.4% |
| Sparse Networks from Scratch [6] | 30% | 75.9% | 92.9% |
| Ours - All Layers Sparse | 30% | 76.9% | 93.4% |
| Ours - First Layer Dense | 30% | 77.1% | 93.5% |
| Sparse Networks from Scratch [6] | 100% | 77.0% | 93.5% |
| Ours - Dense Baseline | 100% | 77.5% | 93.7% |

**Targeted Dropout:** The authors of [10] present a simple and effective method for training a network which is robust to subsequent pruning. Their method outperforms variational dropout [23] and $L_0$ pruning [21]. We compare with *Weight Dropout/Pruning* from [10], which we denote as **TD**. Section B of the Appendix contains more information, experimental details, and hyperparameter trials for the Targeted Dropout experiments, though we provide the best result in Table 2.

**Neural Architecture Search:** As illustrated by Table 3, our network (with a very simple MobileNetV1 like structure) is able to achieve comparable accuracy to an expensive method which performs neural architecture search using reinforcement learning [30].

### 3.4  Training Sparse Neural Networks

We may apply our algorithm for Discovering Neural Wirings to the task of training sparse neural networks. Importantly, our method requires no fine-tuning or retraining to discover a sparse sub-networks – the sparsity is maintained throughout training. This perspective was guided by the the work of Dettmers and Zettelmoyer in [6], though we would like to highlight some differences. Their work enables faster training, though our backwards pass is still dense. Moreover, their work allows for a redistribution of parameters across layers whereas we consider a fixed sparsity per layer.

Our algorithm for training a sparse neural network is similar to Algorithm 1, though we implicitly treat each convolution as a separate graph where each parameter is an edge. For each convolutional layer on the forwards pass, we use the top $k\%$ of the parameters chosen by magnitude. On the backwards pass we allow the gradient to flow to, but not through, all weights that were zeroed out on the forwards pass. All weights receive gradients as if they existed on the forwards pass, regardless of if they were zeroed out.

As in [6] we leave the biases and batchnorm dense. We compare with the result in Appendix C of [6], as we also use a tuned version of a ResNet50 that uses modern optimization techniques such as cosine learning rate scheduling and warmup[5]. We train for 100 epochs and showcase our results in Table 4.

## 4  Conclusion

We present a novel method for discovering neural wirings. With a simple algorithm we demonstrate a significant boost in accuracy over randomly wired networks. We benefit from overparameterization during training even when the resulting model is sparse. Just as in [35], our networks are free from the typical constraints of NAS. This work suggests exciting directions for more complex and efficient methods of discovering neural wirings.

## Acknowledgments

We thank Sarah Pratt, Mark Yatskar and the Beaker team. We also thank Tim Dettmers for his assistance and guidance in the experiments regarding sparse networks. This work is in part supported by DARPA N66001-19-2-4031, NSF IIS-165205, NSF IIS-1637479, NSF IIS-1703166, Sloan Fellowship, NVIDIA Artificial Intelligence Lab, the Allen Institute for Artificial Intelligence, and the AI2 fellowship for AI. Computations on `beaker.org` were supported in part by credits from Google Cloud.

## Footnotes

[1]We follow [36, 22] and define FLOPS as the number of Multiply Adds.

[2]Weight decay [18] may in fact be very helpful for eliminating dead ends.

[3]We use two $3 \times 3$ strided convolutions. The first is standard while the second is depthwise-separable.

[4]We use a 5th order Runge-Kutta method [29] as implemented by [3] (from $t = 0$ to 1 with tolerance 0.001).

[5]We adapt the code from https://github.com/NVIDIA/DeepLearningExamples/tree/master/PyTorch/Classification/RN50v1.5, using the exact same hyperparameters but training for 100 epochs.

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
