[Supplementary Material]

# A  Architecture

Table 5: The general structure of MobileNetV1 vs. MobileNetV1-DNW for ImageNet experiments. `dwconv` denotes *depthwise convolutions*, `pwconv` denotes *pointwise convolution*, `FC` denotes a fully connected layer, $d$ denotes width multiplier [12], $c$ denotes the number of output channels, and $s$ denotes stride. When omitted assume that stride is 1. Batch-norm [14] and ReLU follow each convolution in MobileNetV1. When training on CIFAR-10 [16] the first convolution has stride 1.

| Stage | Output | MobilNetV1 | MobileNetV1-DNW |
|---|---|---|---|
| 0 | $112 \times 112$ | $3 \times 3$ `conv`, $c = 32d$, $s = 2$ | $g_\phi = (3 \times 3$ `conv`, $c = 32$, $s = 2)$ |
| 1 | $112 \times 112$ | $3 \times 3$ `dwconv`, $c = 32d$ <br> $3 \times 3$ `pwconv`, $c = 64d$ | $\mathcal{G}_1$ with $|\mathcal{V}| = 32 + 64$ <br> $|\mathcal{V}_0| = 32$, $|\mathcal{V}_E| = 64$, $|\mathcal{B}| = 2$ |
| 2 | $56 \times 56$ | $3 \times 3$ `dwconv`, $c = 64d$, $s = 2$ <br> $3 \times 3$ `pwconv`, $c = 128d$ <br> $3 \times 3$ `dwconv`, $c = 128d$ <br> $3 \times 3$ `pwconv`, $c = 128d$ | $\mathcal{G}_2$ with $|\mathcal{V}| = 64 + 2 * 128$ <br> $|\mathcal{V}_0| = 64$, $|\mathcal{V}_E| = 128$, $|\mathcal{B}| = 3$ |
| 3 | $28 \times 28$ | $3 \times 3$ `dwconv`, $c = 128d$, $s = 2$ <br> $3 \times 3$ `pwconv`, $c = 256d$ <br> $3 \times 3$ `dwconv`, $c = 256d$ <br> $3 \times 3$ `pwconv`, $c = 256d$ | $\mathcal{G}_3$ with $|\mathcal{V}| = 128 + 2 * 256$ <br> $|\mathcal{V}_0| = 128$, $|\mathcal{V}_E| = 256$, $|\mathcal{B}| = 3$ |
| 4 | $14 \times 14$ | $3 \times 3$ `dwconv`, $c = 256d$, $s = 2$ <br> $3 \times 3$ `pwconv`, $c = 512d$ <br><br> $5 \times \begin{cases} 3 \times 3 \text{ dwconv}, c = 512d \\ 3 \times 3 \text{ pwconv}, c = 512d \end{cases}$ | $\mathcal{G}_4$ with $|\mathcal{V}| = 256 + 6 * 512$ <br> $|\mathcal{V}_0| = 256$, $|\mathcal{V}_E| = 512$, $|\mathcal{B}| = 7$ |
| 5 | $7 \times 7$ | $3 \times 3$ `dwconv`, $c = 512d$, $s = 2$ <br> $3 \times 3$ `pwconv`, $c = 1024d$ <br> $3 \times 3$ `dwconv`, $c = 1024d$ <br> $3 \times 3$ `pwconv`, $c = 1024d$ | $\mathcal{G}_5$ with $|\mathcal{V}| = 512 + 2 * 1024$ <br> $|\mathcal{V}_0| = 512$, $|\mathcal{V}_E| = 1024$, $|\mathcal{B}| = 3$ |
| 6 | 1000 | $7 \times 7$ `pool`, $1024d \times 1000$ `FC` | $h_\psi = (7 \times 7$ `pool`, $1024 \times 1000$ `FC`)$ |

# B  Targeted Dropout: Details, Regular vs. Unconstrained and Additional Hyperparameters

The method of Targeted Dropout is as follows: choose the bottom $\gamma$ fraction of weights by magnitude and apply dropout with probability $\rho$. We use the same architecture as MobileNetV1-DNW ($\times 0.225$) and fix $\gamma$ at each stage so that the network *post pruning* has the same number of edges per stage as MobileNetV1-DNW ($\times 0.225$).

In the setting we consider, *unconstrained* targeted weight dropout outperforms the targeted weight dropout presented in [10] (which we refer to as *regular*). Accordingly, the results we present in Table 2 correspond to unconstrained targeted *weight dropout*. In *regular* targeted weight dropout, dropout is applied to the bottom $\gamma$ fraction of incoming weights *to each neuron*. In *unconstrained* targeted weight dropout, we apply dropout to the bottom $\gamma$ fraction of edges at a given spatial resolution. Accordingly, neurons may die and have no incoming or outgoing edges. We compare unconstrained and regular targeted dropout in Table 6.

Table 6: Comparing variants of targeted weight dropout using the architecture described in Table 5 and tested on CIFAR-10 shown as mean and std over 5 runs.

| Model | Accuracy (Unconstrained) | Accuracy (Regular) |
|---|---|---|
| TD $\rho = 0.9$ | $89.0 \pm 0.2\%$ | $87.9 \pm 0.5\%$ |
| TD $\rho = 0.95$ | $\mathbf{89.2 \pm 0.4\%}$ | $87.9 \pm 0.2\%$ |
| TD $\rho = 0.99$ | $88.6 \pm 0.2\%$ | $87.7 \pm 0.3\%$ |
| TD $\rho = \gamma$ | $88.8 \pm 0.2\%$ | $87.9 \pm 0.2\%$ |

## C   A More General Case

We now consider the case where the *hallucinated edge* $(i, \ell)$ replaces $(j, k) \in \mathcal{E}$.

As before we use $\tilde{w}$ to denote the weight $w$ after the weight update rule $\tilde{w}_{uv} = w_{uv} + \left\langle Z_u, -\alpha \frac{\partial \mathcal{L}}{\partial \mathcal{I}_v} \right\rangle$. We assume that $\alpha$ is small enough so that $\text{sign}(\tilde{w}) = \text{sign}(w)$.

**Claim:** Assume $\mathcal{L}$ is Lipschitz continuous. There exists a learning rate $\alpha^* > 0$ such that for $\alpha \in (0, \alpha^*)$ the process of swapping $(i, \ell)$ for $(j, k)$ will decrease the loss when the state of the nodes are fixed, there is no path from $i$ to $j$, and $|w_{i\ell}| < |w_{jk}|$ but $|\tilde{w}_{i\ell}| > |\tilde{w}_{jk}|$.

*Proof.* Let $\mathcal{A}_k, \mathcal{A}_\ell$ be value of $\mathcal{I}_k$ and $\mathcal{I}_\ell$ after the update rule if $(j, k)$ is replaced with $(i, \ell)$. Let $\mathcal{B}_k$ and $\mathcal{B}_\ell$ be the state of $\mathcal{I}_k$ and $\mathcal{I}_\ell$ after the update rule if we do not allow for swapping. $\mathcal{A}_k, \mathcal{A}_\ell, \mathcal{B}_k$ and $\mathcal{B}_\ell$ are then given by

$$\mathcal{A}_k = \sum_{(u,k) \in \mathcal{E},\ u \neq j} \tilde{w}_{uk} Z_u, \qquad \mathcal{B}_k = \tilde{w}_{jk} Z_j + \sum_{(u,k) \in \mathcal{E},\ u \neq j} \tilde{w}_{uk} Z_u \qquad (14)$$

$$\mathcal{A}_\ell = \tilde{w}_{i\ell} Z_i + \sum_{(u,\ell) \in \mathcal{E},\ u \neq i} \tilde{w}_{u\ell} Z_u, \qquad \mathcal{B}_\ell = \sum_{(u,\ell) \in \mathcal{E},\ u \neq i} \tilde{w}_{u\ell} Z_u. \qquad (15)$$

Additionally, let $g_k = -\alpha \frac{\partial \mathcal{L}}{\partial \mathcal{I}_k}$ and $g_\ell = -\alpha \frac{\partial \mathcal{L}}{\partial \mathcal{I}_\ell}$ be the direction in which the loss most steeply descends with respect to $\mathcal{I}_k$ and $\mathcal{I}_\ell$. By *Lemma 1* (Section D of the Appendix) it suffices to show that

$$\langle \mathcal{A}_k - \mathcal{I}_k, g_k \rangle + \langle \mathcal{A}_\ell - \mathcal{I}_\ell, g_\ell \rangle \geq \langle \mathcal{B}_k - \mathcal{I}_k, g_k \rangle + \langle \mathcal{B}_\ell - \mathcal{I}_\ell, g_\ell \rangle \qquad (16)$$

which simplifies to

$$\tilde{w}_{i\ell} \langle Z_i, g_\ell \rangle \geq \tilde{w}_{jk} \langle Z_j, g_k \rangle \qquad (17)$$

$$\iff \tilde{w}_{i\ell}(\tilde{w}_{i\ell} - w_{i\ell}) \geq \tilde{w}_{jk}(\tilde{w}_{jk} - w_{jk}). \qquad (18)$$

We are now in the equivalent setting as equation 6 and may complete the proof as before.

In practice there may be a path from $i$ and $j$ the state of the nodes will never be the fixed due to stochasticity of mini-batches and updates to the rest of the parameters in the network. However, as the graph grows large the state of one node will have little effect on the state of another, even if there is a path between them. The proofs are done in an idealized case and the empirical results demonstrate that the method works in practice.

## D   Lemma 1

Here we show that for sufficiently small $\alpha$,

$$\left\langle \gamma_1, -\alpha \frac{\partial \mathcal{L}}{\partial \mathcal{I}_v} \right\rangle > \left\langle \gamma_2, -\alpha \frac{\partial \mathcal{L}}{\partial \mathcal{I}_v} \right\rangle \qquad (19)$$

implies that

$$\mathcal{L}\left(\mathcal{I}_v + \alpha \gamma_1\right) < \mathcal{L}\left(\mathcal{I}_v + \alpha \gamma_2\right). \qquad (20)$$

Note that for brevity we have written the loss as a function of $\mathcal{I}_v$. By taking a Taylor expansion we find that

$$\mathcal{L}\left(\mathcal{I}_v + \alpha\gamma\right) \tag{21}$$

$$= \mathcal{L}\left(\mathcal{I}_v\right) + \left\langle \alpha\gamma, \frac{\partial \mathcal{L}}{\partial \mathcal{I}_v} \right\rangle + \mathcal{O}(\alpha^2) \tag{22}$$

$$\tag{23}$$

and so for sufficiently small $\alpha$

$$\mathcal{L}\left(\mathcal{I}_v\right) - \mathcal{L}\left(\mathcal{I}_v + \alpha\gamma\right) \approx \left\langle \gamma, -\alpha\frac{\partial \mathcal{L}}{\partial \mathcal{I}_v} \right\rangle \tag{24}$$

which completes the lemma.

An equivalent argument holds for two dimensions.

$$\left\langle \gamma_1, -\alpha\frac{\partial \mathcal{L}}{\partial \mathcal{I}_v} \right\rangle + \left\langle \xi_1, -\alpha\frac{\partial \mathcal{L}}{\partial \mathcal{I}_u} \right\rangle > \left\langle \gamma_2, -\alpha\frac{\partial \mathcal{L}}{\partial \mathcal{I}_v} \right\rangle + \left\langle \xi_2, -\alpha\frac{\partial \mathcal{L}}{\partial \mathcal{I}_u} \right\rangle \tag{25}$$

implies that

$$\mathcal{L}\left(\mathcal{I}_v + \alpha\gamma_1, \mathcal{I}_u + \alpha\xi_1\right) < \mathcal{L}\left(\mathcal{I}_v + \alpha\gamma_2, \mathcal{I}_u + \alpha\xi_2\right). \tag{26}$$

By taking a Taylor expansion we find that

$$\mathcal{L}\left(\mathcal{I}_v + \alpha\gamma, \mathcal{I}_u + \alpha\xi\right) \tag{27}$$

$$= \mathcal{L}\left(\mathcal{I}_v, \mathcal{I}_u\right) + \left\langle \alpha\gamma, \frac{\partial \mathcal{L}}{\partial \mathcal{I}_v} \right\rangle + \left\langle \alpha\xi, \frac{\partial \mathcal{L}}{\partial \mathcal{I}_u} \right\rangle + \mathcal{O}(\alpha^2) \tag{28}$$

$$\tag{29}$$

and so for sufficiently small $\alpha$

$$\mathcal{L}\left(\mathcal{I}_v, \mathcal{I}_u\right) - \mathcal{L}\left(\mathcal{I}_v + \alpha\gamma, \mathcal{I}_u + \alpha\xi\right) \approx \left\langle \gamma, -\alpha\frac{\partial \mathcal{L}}{\partial \mathcal{I}_v} \right\rangle + \left\langle \xi, -\alpha\frac{\partial \mathcal{L}}{\partial \mathcal{I}_u} \right\rangle. \tag{30}$$

# E    Effect of $\theta_v$ on $w_{uv}$

One concern is that convolution or batch normalization [14] in $f_{\theta_v}$ would make it difficult to choose edges by magnitude. Consider the case where a convolutional kernel is scaled up arbitrarily in magnitude. Even if the incoming edges were important, their magnitudes would likely be small. As a consequence, they would not be chosen.

Here we argue that this is unlikely to occur. When batch normalization [14] is present, the "energy" of the incoming weights are conserved. A formal treatment of this statement is provided by Thoerem 3 of [31].

# F    Reformulation as a Straight-Through Estimator

We now reformulate the update rule as a straight-through estimator [1, 15]. We are equivalently computing the input to node $v$ as

$$\mathcal{I}_v = \sum_{u \in \mathcal{V}} h(w_{uv})Z_u \tag{31}$$

where $h(w_{uv}) = w_{uv}\mathbb{1}_{\{|w_{uv}| > \tau\}}$ in the forward pass. Even though $h$ has gradient 0 when $|w_{uv}| \leq \tau$, we would still like a mechanism for updating $w_{uv}$ in the backward pass. Here we may use the "straight-through" estimator [1], and let $h$ be the identity in the backward pass (*i.e.* we go straight-through $h$). Then $\mathcal{I}_v = \sum_{u \in \mathcal{V}} w_{uv}Z_u$ in the backward pass and we then compute

$$\frac{\partial \mathcal{I}_v}{\partial w_{uv}} = Z_u \tag{32}$$

which, when using the chain rule and a standard SGD update, aligns with our update rule (line 10 in Algorithm 1). This is exactly how we implement the update rule in PyTorch [24].