[Reviews · NeurIPS 2019]

Reviewer 1



This paper explores an important direction in NAS. Different from previous work that builds upon well-designed building blocks, this paper tries to learn connections between channels. Therefore, such methods can potentially find better basic building blocks that are not discovered by human. I really appreciate efforts in this direction. However, regarding the search space proposed in this paper, I suspect such irregular connections between channels are very unfriendly to hardware. Even the FLOPs of the searched model is close to human-designed models, the real hardware efficiency (e.g., latency) may be much worse. Can the authors also include the latency in Table 3 for reference? If it is the case, do authors have any solution to the challenge? The experiment results are promising in very low FLOPs regime. However, results in high FLOPs (200M+, 300M+, 600M, etc.) regime are not provided due to the efficiency issue of the training algorithm. I am very curious about the learned connection patterns. Can the authors provide some visualizations of their learned architectures? Additionally, I find that the source code is not provided. Do the authors have any plan about releasing the code? [Post-Author-Response Update] I am satisfied with the author response. Therefore, I increase the overall score from 6 to 7.

Reviewer 2



The core value of the submission is in the idea and formulation. This is perhaps somewhat subjective, and so hard it is hard to describe the supporting evidence, but it is an especially elegant description of the network design problem. The unification of core parts of the sparse neural network literature combined with the neural architecture search problem is very nice, and will mean that the paper of interest to a very large cross-section of the NeurIPS community. The setup of the comparisons is limited in some ways, but smaller choices were clearly made well. For instance, in the negative, only a few sample networks are presented. However, the authors clearly did make some (admittedly small) effort to present multiple points along the cost/accuracy tradeoff curve for their and comparison methods. The comparison methods include a nice cross-section of near-state-of-the-art networks, though the baseline method the submission builds on for its own method is only MobileNetV1. Overall, the completeness and quality of the experiments are in the range where they are more than sufficient for validating the new and interesting idea that is in the submission, but would be only slightly insufficient for "SOTA" results on a more incremental contribution. One of the most difficult complications that prevent the kind of strategy in the submission from being effective is the handling of stride/resolution/scale between different feature maps. The submission acknowledges around line 135 that this is "not obvious." The description of how this is solved is very cursory, though all of the experiments do mention partial details, including for example Table 4 in the supplement listing describing the striding in comparison and result architectures. However, I am still uncertain on some details. How exactly are different strides included among the possible edges? It is simply the cartesian product of some set of integer strides and the possible set of edges varying on other parameters? Or some subset? The best qualities of the submission are overall a little on the "soft" side: in that it is focused on presenting a formulation/problem/idea and, generally speaking, has relatively little experiments or formal theory. For instance, I suspect there is more justification and substance to some very key design decisions in the the edge update rule in line 5 of Algorithm 1 than is described by the authors, but I do believe the choice is correct even though the reason why is not fully described in the submission. By contrast, though, there maybe need to be more description of how the edge swapping described in section 2 relates to the particular formulation of that step in the proposed method. However, despite this "softness" the idea is very good, and the experiments are at least sufficient for a proof of concept. I'll therefore assert that the submission would be interesting and valuable to NeurIPS proceedings readers if accepted. == Updates after rebuttal/discussion == The authors gave some further explanation of the lower-level details of their method, and these look reasonable. The planned code release will very thoroughly address these and is likely the best approach: since the paper introduces very new methods and concepts it most likely would otherwise be difficult to fully specify in a short conference-paper format. The submission to the MicroNet challenge, that is going to be held with NeurIPS, is also likely to help other researchers in understanding the particulars of the work. My overall score is unchanged, but I am somewhat more confident in recommending acceptance.

Reviewer 3



Comments after rebuttal: I am very satisfied with the rebuttal, especially with the new results (the sparsity results are very interesting), the visualization, and the extended intuition on seeing the method as approximating the backward pass. I have changed my score accordingly. ------------------------------------------ Originality: As far as I am aware, the idea of learning connections in a channel-wise level is novel, as is the method to do this efficiently (instead of accounting for all possible channel - channel connections in the network). However, note that learning connection patterns in neural networks has been done in a layer-wise level in Savarese and Maire, which also includes discrete-time dynamics to some weak extent -- with that in mind, I disagree that the work of Saining et al on exploring randomly connected networks is the most similar approach, as is claimed on line 32. On the other hand, the approach in Savarese and Maire differs from the one proposed here in many ways, including the method to learn the connections and the experimental scope, so I still consider this submission highly original. The work in Zhou et al might also be worth discussing: the discovered supermask is essentially a 'good' wiring for the task, even when the weights are randomly initialized (although the supermask is not truly learned). Finally, the work of Prabhu et al on expanders predates Saining et al and should be mentioned as well. I would be happy to see a more throughout discussion on relevant work. ------------------------------------------ Quality: The method is simple, clean and allows for the discovery of very efficient networks, providing an impressive performance boost when compared to MobileNets in the low-FLOP spectrum. Unfortunately the paper lacks results with larger networks -- I assume because the method is not efficient enough -- which makes it hard to compare against other neural architecture search methods, which discover models more in the 500M FLOP range. I think the impressive results in the low FLOP regime are valuable enough to create interest in learning the connection patterns of neural networks. While the claims help motivate the method, they could be stated a bit more formally, e.g. make it explicit that the loss that decreases after swapping edges is the mini-batch loss, not the full loss (over the whole dataset), and maybe add a footnote arguing why \alpha^\star (line 122) actually exists (or even show it explicitly, as it would only take one line). ------------------------------------------ Clarity: The paper is mostly clear in its writing, but some parts could be better detailed/explained. For example, there is little discussion on the small scale experiments on CIFAR (section 3.1): a model description on the static, discrete and continuous networks would be useful, including some discussion on why the discrete network outperforms the static one (since they both have 41k parameters, are they trained with the same number of nodes, how does k differ, etc). I also personally feel that figures 1 and 2 don't add enough value compared to the space they take: both focus on the static/dynamic graph relationship, which is not a main part of the paper (the convincing results are on static graphs only) -- the same goes for section 2.3: in general, while I think learning connections of dynamic graphs is interesting, it is poorly explored experimentally in the paper, so I don't see why allocate so much space and discussion to it. It would also be worth discussing the relation between the a connection weights {w_uv}_u and \theta_v. For some node v that performs 3x3 conv on its input I_v, the kernel can be scaled up arbitrarily while scaling down all the weights w_uv, which would not change Z_v -- however, this process would place edges (u->v) in the hallucinated set, as they can be scaled down arbitrarily close to zero. For the case where the node v computes BN->ReLU->Conv, scaling down all weights w_uv by the same value should not change Z_v due to batch normalization. This invariance can be problematic since the proposed method does not capture it. Likely typos: - I believe in line 122, the definition of \alpha^\star should have w_{i,k} instead of \hat w_{i,k}. - Equation 2 should have Z_u instead of Z_v ------------------------------------------ Significance: Although the results are not extensive and focus on a specific setting (low FLOP range of mobile-like nets), they are impressive and improve upon very efficient networks, and perform similarly to MNASNet. A comparison against other NAS-based methods is not truly possible since the approaches discover networks with very different FLOP ranges. I believe the paper has a strong significance, as it studies the still poorly explored venue of learning good connection patterns for neural networks, and improves upon the state-of-the-art (again, in a restricted setting). The method to perform this efficiently can be applied to other settings (even more fine-grained than channel to channel connections), so it might have relevance for other works. References: [1] Pedro Savarese, Michael Maire - Learning Implicitly Recurrent CNNs Through Parameter Sharing [2] Hattie Zhou, Janice Lan, Rosanne Liu, Jason Yosinski - Deconstructing Lottery Tickets: Zeros, Signs, and the Supermask [3] Ameya Prabhu, Girish Varma, Anoop Namboodiri - Deep Expander Networks: Efficient Deep Networks from Graph Theory

[Author Response · NeurIPS 2019]

We are very grateful to the reviewers for their insightful, thorough, and thoughtful reviews.

• **R1 & R2**: **Code release.** We will release code, pretrained models and training config files so
that the work is reproducible. We will also include additional low-level details in the paper.

• **R1**: **Visualization.** With thousands of nodes it is difficult to visualize connectivity in one figure
though we are working on an interactive visualization in d3. See right for progress screen-shots
of a smaller net shown early (top) and slightly later (bottom) in training (viewed best with zoom).

• **R1**: **Hardware.** The models are indeed inferior in terms of latency on current hardware. As per
your suggestion we will update the paper to include this detail in addition to our response: Our key
points are that **1)** We agree with the philosophy of the NeurIPS 2019 MicroNet Challenge – it may
be important to explore models which might guide the direction of new hardware development
(we are planning submission to the challenge). **2)** We will release a version of our model which
performs inference via the Deep Graph Library (dgl) in Pytorch, we hope that dgl and similar libraries will be faster in
the years to come. **3)** We also expect fast inference via sparse GPU kernels, which we anticipate will be developed.

• **R2**: **Sparsity:** Thank you for mentioning that *the unification of core parts of the sparse neural network literature*
*combined with the neural architecture search problem is very nice*. As a useful experiment to include to showcase
this, we apply Algorithm 1 separately to each filter of the version of ResNet50 used in [1] and obtain state of the
art performance at the task of *training sparse networks from scratch* as described in [1]. With $10\%$ and $20\%$ of the
parameters we achieve **74.6**% and **76.6**% top-1 accuracy respectively on ImageNet (keeping a dense first-layer – less
than 10k params – we achieve **75.5**% and **76.8**%). SOTA in [1] is 73.1% for 10% and 74.9% for 20%.

• **R2**: **Resolution & Stride:** As we discuss briefly in section 2.4, for the large scale experiments we consider a separate
graph for each spatial resolution (the output of graph $\mathcal{G}_i$ is given as input to graph $\mathcal{G}_{i+1}$) and the convs performed at the
input nodes are strided. The number of output nodes is the channels that MobileNetV1 has for that resolution. As we
will elaborate on in the final revision, in the small scale setting either stride 1 may be used everywhere or some nodes
may perform strided convs followed by zero-padding (as in Figure 2 of the paper).

• **R2**: **Theoretical Claims:** We agree that this work will benefit from theoretical analysis which extends beyond
decreasing the loss on the mini-batch with the SGD update. In the final revision we believe that it will be useful to
elaborate on this work from the perspective of the Lottery Ticket hypothesis and the myriad of recent work that seeks to
explain it (wherein weights whose magnitude tends to zero are less important, as in line 5 of Algorithm 1) such as [2].
For now we invite you to reference the Extended Intuition section below for additional justification and in the final
revision we will attempt to discuss or at least conjecture the extension of [2] to a student and teacher wiring.

• **R3**: **Extended Intuition.** We thank you for the excellent suggestion of offering the additional motivation of
performing an approximation in the backwards pass. We are happy to see that we may arrive at the same objec-
tive in this way and we will include this exciting result. Indeed, we are calculating $\mathcal{I}_v = \sum_{u \in \mathcal{V}} g(w_{uv}) w_{uv} Z_u$
where $g(w_{uv}) = \mathbb{1}_{\{|w_{uv}| > \tau\}}$ in the forwards pass. Our method offers the following interpretation – we are
using a soft*er* version of $g(w_{uv})$ in the backwards pass where $g$ approaches 1 for large magnitude weights.
Ignoring weights with $|w_{uv}| < \epsilon \ll \tau$ we may let $g(w_{uv}) = 1 - \epsilon|w_{uv}|^{-1} = 1 - \epsilon w_{uv}^{-1}\text{sign}(w_{uv})$
where $\text{sign}(w_{uv})$ is 1 if $w_{uv} > 0$ and $-1$ otherwise. Aligning with the update rule as needed,
$\frac{\partial \mathcal{I}_v}{\partial w_{uv}} = \frac{\partial g(w_{uv})}{\partial w_{uv}} w_{uv} Z_u + g(w_{uv}) Z_u = \epsilon w_{uv}^{-2}\text{sign}(w_{uv}) w_{uv} Z_u + Z_u - \epsilon w_{uv}^{-1}\text{sign}(w_{uv}) Z_u = Z_u.$

• **R3**: **Related work.** We agree that this work would benefit from a better discussion of related
material and less dynamic material. Thank you for the relevant papers which we will include.

• **R3**: **Clarity.** We will include more details concerning the small-scale experiment, wherein each node performs
normalization followed by conv and relu and in all cases the graphs have 41,000 parameters. The experiments showcase
that in the setting we consider, dynamic graphs are more expressive. We will also discuss the important relation between
$w$ and $\theta$, including the effects of kernel scaling and batchnorm in which we believe theorem 3 from [2] may be crucial.

• **R3**: **500M FLOP Regime.** We explore the very low FLOP regime as it is an interesting application of the method
– with enough wiring, a good wiring is likely less essential. During this response period we have completed a quick
experiment to showcase that our method also performs well in larger FLOP settings. Since we are limited in time we
only train for 100 epochs (250 epochs is standard for this setting). We take MobileNetV1 (width-mult 2) and restrict the
number of edges to be the same as MobileNetV1 (width-mult 1) then apply our method to achieve **74.4**% accuracy
on ImageNet with 582 MFLOPs and 4.2M params. NASNet-A, PNAS, DARTS, and vanilla MobileNetV1 achieve
respectively 74.0, 74.2, 73.1, and 70.6 with 564, 588, 595, and 569 MFLOPs and 5.3M, 5.1M, 4.9M, and 4.2M params.

[1] T. Dettmers and L. Zettlemoyer, "Sparse networks from scratch: Faster training without losing performance," 2019.
[2] Y. Tian, T. Jiang, Q. Gong, and A. Morcos, "Luck matters: Understanding training dynamics of deep relu networks," 2019.

$\tau = 0.5, \epsilon = 0.05$

[Meta-Review · NeurIPS 2019]

The paper presents an interesting method to efficiently learn channel-wise wirings in deep neural networks, with compelling results in the low FLOPS regime. It has practical value and should be of wide interest to the NeurIPS community. All reviewers recommend acceptance, and the AC agrees with this decision.